

# Holocene hydroclimate reconstruction based on pollen, XRF, and grain-size analysis and its implications for past societies of the Korean Peninsula

Jinheum Park[1], Jungjae Park[1,2], Sangheon Yi[3,4], Jaesoo Lim[3], Jin Cheul Kim[3], Qiuhong Jin[1], Jieun Choi[1]

[1]Department of Geography, Seoul National University, 1, Gwanak-ro, Gwanak-gu, Seoul, 08826, Republic of Korea
[2]Institute for Korean Regional Studies, Seoul National University, 1, Gwanak-ro, Gwanak-gu, Seoul, 08826, Republic of Korea
[3]Geology Division, Korea Institute of Geoscience and Mineral Resources, 124, Gwahak-ro, Yuseong-gu, Daejon, 34132, Republic of Korea
[4]Department of Petroleum Resources Technology, University of Science and Technology, 217, Gajeong-ro, Yuseong-gu, Daejeon, 34113, Republic of Korea

*Correspondence to*: Jungjae Park (jungjaep@snu.ac.kr) and Sangheon Yi (shyi@kigam.re.kr)

**Abstract.** The dynamics of the East Asian Summer Monsoon (EASM) and their link to past societies during the Holocene are topics of growing interest. In this study, we present analyses of a ca. 6,000-year pollen record, as well as X-ray
fluorescence (XRF) and grain-size data from the STP18-03 core sampled from Miryang in the Korean Peninsula, which spans ca. 8.3–2.3 ka BP. In-phase relationships of these proxies revealed an imprint of the Holocene Climate Optimum (HCO) during the early to mid-Holocene and subsequent drying toward the late Holocene in accordance with decreasing solar insolation. At centennial timescales, our study indicates wet conditions during ca. 8.3–7.5, 7.1–6.4, 6.0–4.8, and 3.6–2.8 ka BP, and a drier climate during ca. 7.5–7.1, 6.4–6.0, and 4.8–3.6 ka BP. Notably, our finding for ca. 6.4–6.0 ka BP
contributes further evidence of a drying event in the Korean Peninsula during this period. We suggest that the Pacific Ocean played a role in the underlying mechanism of hydroclimate change in the region. A strong Kuroshio Current (KC) and long-term El Niño–Southern Oscillation (ENSO)-like variability in the Western Tropical Pacific (WTP) were closely linked to the influence of the EASM over the Korean Peninsula. In particular, dry phases during ca. 4.8–3.6 and 2.8–2.3 ka BP, which were synchronous with a more active ENSO, closely corresponded to lower population levels according to a summed
probability distribution (SPD) of archaeological records assembled in the Korean Peninsula. This finding implies that past human societies of Korea were highly vulnerable to climate deterioration caused by precipitation deficits.

## 1 Introduction

The East Asian Summer Monsoon (EASM) is part of the global monsoon system, and is a major driver of the East Asian Holocene climate (An, 2000; Dykoski et al., 2005; Ji et al., 2005; Wang et al., 2005; Chen et al., 2015a; Stebich et al., 2015).
The temporally and spatially complex nature of the EASM (An et al., 2000; Chen et al., 2015b; Zhou et al., 2016) requires further high-resolution studies from many different sites within the region to better reconstruct its impact throughout the



Holocene. For example, although many paleoclimate studies have focused on mainland China as a key region representing East Asia, site-specific and varying imprints of the EASM within the region are being increasingly addressed between the interior and coastal parts of China (Chen et al., 2015b; Zhou et al., 2016; Zhu et al., 2017) as well as between China and

Korea (Park et al., 2017). In the Korean Peninsula, the EASM has been suggested to be primarily driven by the Pacific Ocean, in processes such as the Kuroshio Current (KC) flowing from the Western Tropical Pacific (WTP) and the El Niño–Southern Oscillation (ENSO) in the equatorial Pacific (Lim and Fujiki, 2011; Park et al., 2019; Park et al., 2016; Lee et al., 2020). Therefore, more high-resolution studies are needed to examine EASM dynamics closely in relation to oceanic forcing. However, despite increasing research efforts, high-resolution studies of the EASM in the Korean Peninsula remain scarce.

Although pollen analysis has been the most common approach to reconstruct past precipitation changes (Lim and Fujiki, 2011; Park et al., 2019; Lee et al., 2020; Park et al., 2016), this methodology contains physical limitations in enhancing the temporal resolution between samples. In contrast, speleothem-based Holocene EASM reconstructions, which permit finer resolution, have been confined to the last few thousand years in the Korean Peninsula and have not defined a clear link with the Pacific Ocean (Jo et al., 2017; Hong et al., 2012; Yu et al., 2016). Additionally, oxygen isotope values in speleothems

sometimes do not purely reflect rainfall signals (Chen et al., 2015a; Maher and Thompson, 2012; Caley et al., 2014; Lachniet, 2009). Therefore, there is a significant need to assemble multiple types of proxies and maintain high temporal resolution for more accurate analysis of the link between the EASM and oceanic forcing.

Another topic of rising interest is the relationship between climate change and human societies. This issue has been

addressed by many studies worldwide over the past few decades, in regions including Europe (Tallavaara et al., 2015; Büntgen et al., 2011), Greenland (D'Andrea et al., 2011), America (Munoz et al., 2010), Mesopotamia (Weiss et al., 1993; Carolin et al., 2019), India (Dixit et al., 2018), Egypt (Manning et al., 2017), and China (An et al., 2005; Xie et al., 2013; Wang et al., 2014). Most of these studies have supported a dominant climatic role in the vicissitudes of past societies, whereas others have recently reported human resilience to environmental stresses (Flohr et al., 2016; Blockley et al., 2018).

Thus, the relationship between climate and human societies is not always simplistically determined and may be case-specific. Even within sub-continents of China—e.g., Northwest, North, and South China, the Qinghai Tibetan Plateau—and Japan, human activity patterns during the Holocene appear to be distinct (Wang et al., 2014; Crema et al., 2016). It remains unclear whether this discrepancy reflects local differences in climate imprints or differences in societal responses to climate. Therefore, to analyze socio-environmental relationships in a particular area accurately, a direct comparison of continuous,

local, and high-resolution paleoclimate proxy data with human activity indicators is crucial. However, in the Korean Peninsula, this work remains in the nascent stage (Constantine et al., 2019; Park et al., 2019). The topic of climate–society relationships in Korea often depends on individual historical records of natural disasters, leading to assumptive storytelling about their potential impact on contemporary societal unrest (Cho, 2009). For older periods with scarce disaster records, qualitative and broad climate interpretations tend to be expressed as "warm/cold" or "wet/dry", often in comparison with



temporal and spatial patterns of archaeological findings (Ahn and Hwang, 2015), due to a lack of reliable high-resolution
      climate data.

      Therefore, in the present study, we performed a multi-proxy analysis of Holocene hydroclimate change in the Miryang
      region of the Korean Peninsula from ca. 8.3 to 2.3 ka BP. We combined pollen, X-ray fluorescence (XRF), and grain-size

data to reconstruct the EASM history over the Korean Peninsula. The high temporal resolution of our XRF analysis allowed
      us to analyze the paleoclimate dataset at decadal to annual scales. We examined the role of the Pacific Ocean as an
      underlying mechanism for EASM history in the Korean Peninsula, particularly in terms of ENSO variance and heat and
      moisture supply along the KC. We also identified a synchronous change between hydroclimate and past population in the
      Korean Peninsula and explored the processes through which past human societies may have responded to climate impacts

during the Holocene.

## 2 Regional Setting

      Our coring site (35°26'18.84" N, 128°46'41.26" E; 4.64 m a.s.l.) is located in a floodplain of the Miryang River in the
      southeastern part of the Korean Peninsula (Fig. 1a–c). The Miryang River is a tributary of the Nakdong River, which
      eventually flows out to the Korea Strait, between the Korean Peninsula and the Japanese Archipelago. The backdrop of the

coring site is a group of small mountains several hundred meters in elevation, including Jongnamsan (663 m) and
      Palbongsan (391 m) (Fig. 1c). The Korean Peninsula is situated amid the East Asian monsoonal system, and experiences
      seasonal differences in air pressure due to its location between the Eurasia continent and the Pacific Ocean (An, 2000) (Fig.
      1a). In summer, hot and humid southeasterly wind flows into the Korean Peninsula under the influence of the warm KC,
      whereas in winter, cold and dry northwesterly wind from the Siberian High dominates. According to a 30-year (1981–2010

AD) climate record from a meteorological observation station 6 km north of the coring site, the mean annual temperature is
      13.3 °C, with a January minimum of 0.0 °C and August maximum of 25.8 °C. The mean annual precipitation is 1229.4 mm;
      precipitation is lowest in December (16.4 mm) and highest in July (269.5 mm) (Fig. 1d). A total precipitation of 829.5 mm
      during June–September, JJAS) accounts for 67.5 % of the total annual precipitation, indicating significant influence of the
      EASM on the hydroclimate of the region (Korea Meteorological Administration, 2020).


      The regional vegetation of Miryang is composed of temperate broadleaf and mixed forest. On Mt Jongnamsan, arboreal
      species such as *Pinus densiflora*, *Quercus serrata*, *Rhus sylvestris*, *Q. acutissima*, *Castanea crenata*, *Indigofera kirilowii*,
      and *Q. variabilis* and herb species such as *Oplismenus undulatifolius*, *Carex lanceolata*, *Artemisia keiskeana*, *Spodiopogon
      sibiricus*, *Cocculus trilobus*, and *Arundinella hirta* dominate at 30–298 m altitude (Korea National Insitute of Ecology, 2016).

In wetlands along the Miryang River, *Phragmites communis*, *Salix gracilistyla*, *Phragmites japonica*, *Persicaria longiseta*,



*Persicaria thunbergii*, *Juncus effuses var. decipiens*, and *Zizania latifolia* are also found (Ministry of Environment of the Republic of Korea, 2002).

## 3 Materials and Methods

### 3.1 Core materials and dating

In April 2018, the 20-m STP18-03 core was collected in 1-m sections from a floodplain of the Miryang River (Fig. 1). Depth zones of the uppermost 0–1.25 m and lowermost 14–20 m were excluded from all analyses as the former were regarded to have been affected by human activities and the latter consisted mainly of gravel. We sent a total of 16 samples to the Korea Institute of Geoscience and Mineral Resources (KIGAM), Republic of Korea (Table 1) for age dating; eight were measured using the optically stimulated luminescence (OSL) dating method. These samples were treated with $Na_4P_2O_7$, HCl, $H_2O_2$,

and $H_2SiF_6$ to extract quartz with a diameter of 4–11 μm. Then, OSL signals were measured using a TL-DA-20 reader (Risø DTU, Roskilde, Denmark) equipped with a blue light-emitting diode (LED; 470 ± 20 nm) stimulation source. Plant fragments from the other eight samples were used for radiocarbon dating by accelerator mass spectrometry. Based on the results, we constructed an age model using the *bacon* ver. 2.3 R package (Blaauw and Christen, 2011) with the IntCal13 calibration dataset (Reimer et al., 2013) (Fig. 2b). Two radiocarbon dates from depths of 795 cm and 1032 cm were omitted

due to their anomalous ages in relation to those of other samples.

### 3.2 Proxy Analyses

We performed palynological analysis of a total of 137 samples at intervals ranging from 1 to 70 cm (1–6 cm for sections from 395 to 1010 cm and 9–70 cm for the remaining sections). The average temporal resolution was 41.4 years, with minimum and maximum values of 12 and 348 years, respectively. For sample preparation, we followed the standard protocol

of Faegri et al. (1989) including HCl (10 %), KOH (10 %), HF (40 %), and acetolysis. KOH treatment was repeated twice to remove organic matter completely. For highly humic samples in the range of 501–528 cm, the KOH procedure was repeated up to three times. For each sample, one *lycopodium* tablet containing 177,745 spores was added, and at least 300 pollen grains and spores were counted on each slide using a Leica microscope at 400 × magnification. For palynomorph identification, we referred to a pollen atlas from Lake Suigetsu, Japan (Demske et al., 2013). The percentage of pollen was

calculated for each taxon relative to the total sum of non-aquatic pollen grains and spores in the sample. The result was visualized using the Tilia software ver. 2.0.41 (Grimm, 1991), and stratigraphically constrained cluster analysis was conducted using CONISS (Grimm, 1987) based on non-aquatic taxa.

Grain-size analysis was performed using a Mastersizer 2000 laser diffraction particle-size analyzer (Malvern Instruments,

UK) at KIGAM. Approximately 300 mg of each sample was collected at 10-cm intervals from the 205–1390 cm section. These subsamples were treated with $H_2O_2$ (35 %) and HCl (1 N) to remove organic matter and carbonates. Grain sizes of < 4





μm, 4–63 μm, and > 63 μm were classified as clay, silt, and sand, respectively. XRF analysis was also conducted at KIGAM using an XRF core scanner (Avaatech B.V.; Alkmaar, Netherlands), which extracts elemental concentration data in a nearly continuous manner (Croudace et al., 2006; Löwemark et al., 2011). XRF signals were measured from split core surfaces
from depths of 12.5–1293.5 cm, with settings of 10/50 kV and 0.25/1.0 mA, and a sampling time of 30 s. In total, 2,088 values were collected at a resolution of 0.5 cm. However, we did not perform paleoenvironmental interpretations on data at depths above 365 cm, because these sections correspond to periods later than ca. 2.3 ka BP, for which there is evidence of agriculture in the Miryang region (Yoon et al., 2005)

Cross-spectral analysis on the proxy data was conducted using the REDFIT-X software ver. 1.1 (Ólafsdóttir et al., 2016) with 1,000 Monte Carlo simulations (nsim = 1000), an oversampling value of 4.0 for Lomb–Scargle Fourier transform (ofac = 4.0), four segments with 50 % overlap (n50 = 4), and a Welch window (iwin = 1). To avoid statistical bias, we eliminated two pollen values at depths of 380 and 395 cm, which reflected an abrupt shift immediately before cessation of pollen deposition. For all other proxies, we used only values that were within the timespan of the pollen data for analysis.

## 4 Results

### 4.1 Chronology

The chronology of the STP18-03 core contained a record of ca. 6420 years, from the mid- to late Holocene, representing the period from ca. 8340 cal yr BP (1280 cm depth) to ca. 1920 cal yr BP (350 cm depth) (Fig. 2b). The sedimentation rate was 0.38 cm per year for the 790–1280-cm segment and 0.09 cm per year for the 380–790-cm segment (average, 0.14 cm per
year). Sedimentation was continuous throughout the core, except for the 901–905 and 1100–1140-cm segments, perhaps due to on-site technical problems while coring or disturbance by underground water.

### 4.2 Stratigraphy and multi-proxy environmental data

#### 4.2.1 Zone 1 (790–1390 cm)

This zone consists mainly of dark brown clay, although sand is frequently observed in multiple layers (Fig. 2a). Sand
percentages fluctuate significantly between 0.5 % and 79.1 % (Fig. 2c). The amount of titanium (Ti) also varies, with a large amplitude generally in the opposite direction to that of sand content (Fig. 2d). The tree pollen percentage is generally low at 1145–1280 cm, with a minimum value of 77.2 % (Fig. 2e). *Artemisia* (mugwort), *Poaceae* (wild grass), and fern species comprise 20.6 % of the pollen composition (Fig. 2f); this trend is reversed at 790–1090 cm, where the proportion of arboreal pollen remains high and stable at 87.1–96.4 %, and that of *Artemisia* and *Poaceae* pollen together with fern spores remains
low, at an average of 8.7 %. The largest proportion of arboreal pollen is constituted by *Quercus* (oak) (Fig. 3); other broadleaf tree genera include *Alnus* (alder), *Fraxinus* (ash), and *Ulmus* (elm), which reach the highest proportion in this zone





of the core. Notably, at a depth of 950 cm, a sudden increase in *Alnus* pollen to 58.0 % (Fig. 3) coincides with abrupt shifts in sand and Ti content (Fig. 2c and d). This abnormal value implies a sudden local disturbance event that perturbed the pre-existing vegetation and gave rise to pioneer species with high environmental tolerance (McVean, 1953; Weng et al., 2004).

### 4.2.2 Zone 2 (400–790 cm)

From 790 to 400 cm, the clay content gradually increases as depth decreases, and the colour changes from light brown to grey (Fig. 2a). Sand percentages and Ti content stabilize, changing in tandem with the pollen data (Fig. 2c–e). Overall, lower sand content is concurrent with a lower proportion of tree pollen and higher Ti values, and vice versa. Between 590 and 635 cm, where the sand percentage temporarily decreases to 28.8%, the Ti content increases sharply as the proportion of tree pollen drops to as low as 83.1 % due to a decline in the main arboreal genera such as *Quercus*, *Pinus*, *Alnus*, *Fraxinus*, and *Ulmus* (Fig. 3). A similar trend is also observed between 420 and 500 cm, where the proportions of sand and arboreal pollen drop to below 10 % and 80 %, respectively. Upland herbs such as *Artemisia* and *Poaceae* occupy this relatively open area, whereas *Pteridium* ferns rise to a maximum value in this zone. This trend is in contrast with the 500–590 and 635–790 cm sections, in which the proportions of sand and tree pollen remain high as Ti XRF values are low. Notably, a high proportion of tree pollen at 500–590 cm is led by a climax in *Quercus* pollen (Fig. 3).

### 4.2.3 Zone 3 (210–400 cm)

This part of the core is characterized by brown clayey silt (Fig. 2a) with low sand content (Fig. 2c) and high Ti values (Fig. 2d). Pollen deposition is interrupted after an explosive increase in upland herbs (mainly *Artemisia* and *Poaceae*), *Cyperaceae* (sedges), and *Polypodiales undiff.* ferns at the beginning of this zone (Figs. 2e, f and 3). Considering the near absence of sand content in this zone (Fig. 2c), this disruption may have been caused by a cessation of water supply from the river to the floodplain due to either climate drying or river route alteration, and the subsequent exposure of the site to air.

## 5 Discussion

### 5.1 Role of our proxy data as paleoclimate indicators

At millennial timescales, our Ti, pollen, and sand content data are consistent with a declining trend of summer solar insolation in the Northern Hemisphere from the mid- to late Holocene (Berger and Loutre, 1991) (Fig. 4a–e). The gradual decrease in the proportion of arboreal pollen (Fig. 4c) reflects cooling and drying associated with a southward migration of the Intertropical Convergence Zone (ITCZ) induced by orbital forcing (Berger and Loutre, 1991; Haug et al., 2001). Our sand proportion data also follow this trend, as fluvial sand discharge by the Miryang River would have weakened due to less precipitation in the late Holocene relative to earlier periods (Fig. 4e). Ti XRF values, in the opposite direction, change in tandem with these two proxies, such that the signal generally increases towards the late Holocene (Fig. 4b). In many studies, Ti has been used as an indicator of terrestrial erosion, although its paleoenvironmental interpretation may vary according to



regional context (Sun et al., 2008; Bakke et al., 2009). In the present study, considering its synchronicity with arboreal pollen proportion (Fig. 4b and c), we interpret Ti as reflecting hydroclimate change in the Korean Peninsula. During wet periods, more tree growth (mainly oak and pine trees, Fig. 3) in nearby hills would have suppressed soil erosion via the anchoring effect of roots, leading to lower Ti XRF values. However, as climate became drier towards the late Holocene, tree replacement by herbs and ferns would have weakened this effect, resulting in greater Ti erosion.

This attribute of Ti data as a climate proxy in relation to vegetation change is further supported by cross-spectral analysis (Ólafsdóttir et al., 2016) (Fig. 5). The analysis of Ti and arboreal pollen data implies high coherency at frequencies of 518, 148, 127, and 104 years (Fig. 5a). A ~500-year frequency is widely detected in East Asian monsoonal regions and has been attributed to oceanic influences such as thermohaline circulation (THC) or the ENSO, possibly modulated by solar activity (Roth and Reijmer, 2005; Dima and Lohmann, 2009; Xu et al., 2014; Xu et al., 2019; Stebich et al., 2015; Park et al., 2019). This link is corroborated by additional analysis with the WTP SST record from the MD98-2176 core (Stott et al., 2004) and hematite-stained grains (HSG) in the North Atlantic (Bond et al., 2001), which commonly show the ~500-year frequency observed in our Ti values and in the tree pollen data (Fig. 5b–c and e–f). Smaller frequencies of ~150, ~130, and ~100 years are likely attributable to a solar origin (Scuderi, 1993; Roth and Reijmer, 2005); these signals have been interpreted in terms of solar modulation on the EASM strength at Qinghai Lake, central China (Ji et al., 2005) and Jeju Island (Park et al., 2017), south of the Korean Peninsula (Fig. 1a and b). Our additional analysis of total solar irradiation (TSI) data (Steinhilber et al., 2009) contributes further evidence of a solar contribution to these cycles (Fig. 5d and g), although ~130- and ~100-year frequencies fail to reach statistically significant levels between tree proportion and TSI (Fig. 5g), possibly due to the lower temporal resolution of pollen analysis relative to XRF scanning.

## 5.2 Climate change in the Korean Peninsula during the Holocene

From ca. 8.3 to 5.4 ka BP, highly sustained proportions of arboreal pollen (Fig. 4c) likely reflect the influence of the Holocene Climate Optimum (HCO) (An et al., 2000; Wanner et al., 2008; Zhou et al., 2016), which resulted in warmer summers in the Korean Peninsula, possibly increasing annual average temperatures by 1–2 °C compared with the pre-industrial period (Renssen et al., 2012). The climate was generally warm and humid, influenced by the northward advance of the EASM (Yang et al., 2015). However, one exception is an abrupt drop in the proportion of tree pollen at ca. 8.2 ka BP, when pollen from herb taxa including mugwort (*Artemisia*) and wild grass (*Poaceae*) suddenly increased (Fig. 3), reflecting the "8.2 ka event", an abrupt global cooling phenomenon (Alley et al., 1997; Veski et al., 2004; Cheng et al., 2009), which corroborates our previous reports of this event in the Korean Peninsula (Park et al., 2018; Park et al., 2019).

At centennial timescales, several periods of wet and dry climate alternate throughout the mid- to late Holocene. Our Ti, pollen, and sand data indicate relatively wet climates for ca. 8.3–7.5, 7.1–6.4, 6.0–4.8, and 3.6–2.8 ka BP, and drier conditions for ca. 7.5–7.1, 6.4–6.0, and 4.8–3.6 ka BP (Fig. 4b–e). A pronounced feature of this multi-centennial-scale





environmental change is a close connection with SST records from the Pacific Ocean (Sun et al., 2005; Stott et al., 2004)
(Fig. 4f and g). Periods of warm and wet climate are accompanied by high SST records obtained for the Okinawa Trough
(Fig. 4f), which is located directly on the path of the KC (Fig. 1a), which is an important contributor to EASM strength in
coastal East Asia, where it is a major transport mechanism of heat and moisture from the WTP (Jian et al., 2000; Zhou et al.,
2009; Lim and Fujiki, 2011; Hu et al., 2015; Park et al., 2016; Constantine et al., 2019; Lee et al., 2020). During ca. 8.3–7.5,
7.1–6.4, 6.0–4.8, and 3.6–2.8 ka BP, greater heat and water vapor supply along the KC would have enabled active
atmospheric convection and stronger EASM influence over the Korean Peninsula, whereas the opposite would have occurred
during ca. 7.5–7.1, 6.4–6.0, and 4.8–3.6 ka BP.

Among these periods, a sign of drying and/or cooling around 6.4–6.0 ka BP (Fig. 4b–e) is consistent with previous findings
at Lake Pomaeho in the central Korean Peninsula (Constantine et al., 2019) (Fig. 1b). This drying signal may have been
muted in our previous study of Gwangyang (Park et al., 2019) (Fig. 1b, GY-1) due to lower temporal resolution (~80 years)
around the sedimentary section. However, evidence of climate deterioration during 6.4–6.0 ka BP is not consistent across
different regions of East Asia. For example, Daihai Lake (Xiao et al., 2004) and Gonghai Lake (Chen et al., 2015a) in North
China and Dongge Cave in South China (Wang et al., 2005) (Fig. 1a) record abrupt shifts toward less precipitation. However,
in Lake Xiaolongwan (Chu et al., 2014; Xu et al., 2019) and Lake Sihailongwan (Stebich et al., 2015) in Northeast China
(Fig. 1a), this signal is not clear. Although this discrepancy may reflect regionally different climate patterns, we cannot rule
out the possibility of inherent bias in the proxy-based reconstructions. For example, although the DA stalagmite in Dongge
Cave (Wang et al., 2005) (Fig. 1) recorded significant drying around 6.4–6.0 ka BP, the D4 stalagmite (Dykoski et al., 2005),
which was obtained from the same cave, does not exhibit clear changes. The reliability of EASM precipitation signals among
Chinese stalagmites has been questioned due to mixed effects of the Indian Summer Monsoon (Fig. 1a, ISM) and/or
hydrologic processes affecting oxygen isotope values (Maher and Thompson, 2012; Chen et al., 2015a; Caley et al., 2014).
Likewise, in pollen records, source area and/or overestimation issues inherent in palynological methodology (Seppä and
Bennett, 2003) can lead to potential bias in blurring climate signals. In this study, we suspect that a methodological issue
explains the smaller amplitude of the changes in pollen proportions during ca. 7.5–7.1 ka BP, whereas the other sedimentary
proxies, XRF and sand percentage data exhibit clearer phase shifts with the Pacific Ocean (Fig. 4b–g). Similarly, in pollen
records from Daihai Lake (Xiao et al., 2004) and Gonghai Lake (Chen et al., 2015a), drying signals during ca. 7.5–7.1 ka BP
are less evident than those during ca. 6.4–6.0 ka BP. Considering this complexity among different sites and time periods,
imprints of EASM weakening during ca. 7.5–7.1 and 6.4–6.0 ka BP are not yet clearly explicable at the regional scale in East
Asia and therefore require further research with abundant high-resolution data across different study sites.


Our proxy data also indicate a pronounced drying trend in the Korean Peninsula during ca. 4.8–3.6 ka BP (Fig. 4b–e), which
is consistent with global findings of significantly decreased precipitation and/or temperature around this period (Bond et al.,
1997; Wang et al., 2005; Wanner et al., 2011; Bond et al., 2001). In East Asia, this climate impact may have been amplified





by long-term ENSO-like variance, which strengthened from the mid-Holocene (Moy et al., 2002; Conroy et al., 2008;
Donders et al., 2008). The ENSO possibly exerts a dampening effect on EASM intensity by affecting low- to mid-latitude
atmospheric patterns (Xu et al., 2019; Feng and Hu, 2014; Hu et al., 2015), evidence of which has been found in coastal East
Asian regions, particularly during El Niño-like phases (Chen et al., 2015b; Park et al., 2017; Park et al., 2016). Active
ENSO-like forcing would frequently have pushed warm seawater pools in the western Pacific farther to the east
(Timmermann et al., 2018), consequently lowering WTP SST values (Fig. 4g) and weakening the KC (Hu et al., 2015) (Fig.
4f).

### 5.3 Response of past societies to mid- to late Holocene climate change

From ca. 6 ka BP to 2.3 ka BP, the Ti value of the STP18-03 core closely follows the summed probability distribution (SPD)
plots of radiocarbon dates collected from archaeological samples in the Korean Peninsula (Oh et al., 2017) (Fig. 6a). The
SPD method is increasingly used as a proxy for past population levels (Wang et al., 2014; Ahn and Hwang, 2015; Tallavaara
et al., 2015; Crema et al., 2016; Bevan et al., 2017; Oh et al., 2017; Xu et al., 2019) by assembling radiocarbon age
calculations from archaeological findings (Gamble et al., 2005; Surovell et al., 2009). In our data, Ti content gradually
decreased and remained low until ca. 4.8 ka BP, recovered high values by ca 3.6 ka BP, and then diminished again before
increasing at ca. 2.8 ka BP (Fig. 6b). This trend is coherent with, but opposite to, changes in the SPD data, which indicate
larger populations during ca. 6.0–4.8 ka BP and ca. 3.6–2.8 ka BP, and lower populations during ca. 4.8–3.6 ka BP and after
ca. 2.8 ka BP (Fig. 6a). Three abrupt transition points around 4.8, 4.2, and 4.0 ka BP are found in both datasets, validating
their link (Fig. 6a and b). Together with our previous research (Constantine et al., 2019; Park et al., 2019), this robust
synchronicity between Ti data and archaeological records contributes to accumulating evidence that past societies of the
Korean Peninsula responded strongly and with great sensitivity to climate change.

Given its synchronicity with arboreal pollen data, we used the Ti XRF signal as a proxy of climate change and identified two
periods with a relatively warm and wet climate, when past populations increased: ca. 6–4.8 ka BP and ca. 3.6–2.8 ka BP (Fig.
6). This link is accompanied by high SSTs in the Okinawa Trough (Fig. 4f). Notably, the former period corresponds to the
latter part of the HCO, characterized by a warm and wet climate during the mid-Holocene both globally (Wanner et al., 2008;
Renssen et al., 2012; Renssen et al., 2009) and in the Korean Peninsula (Park et al., 2019). During this period, favorable
climate conditions would have enabled successful hunting and gathering, with sufficiently abundant food resources to sustain
population growth (Wang et al., 2014). Besides hunting and gathering, evidence of foxtail, broomcorn, and legume
cultivation has been found at Korean Middle-Late Chulmun (Neolithic) sites for as early as ca. 5.5 ka BP (Lee, 2011).
Although the degree to which farming was an important food source for the Chulmun people remains unclear, it is likely that
a warm and humid climate provided better conditions for successful agriculture (Wang et al., 2014; Xu et al., 2019).
Moreover, the latter period of ca. 3.6–2.8 ka BP was a time of explosive increase in Bronze Age settlements in the Korean
Peninsula, beginning as early as ca. 3.9 ka BP (Kim and Bae, 2010). Similarly, Lee (2011) suggested a major transition from



Chulmun to Mulmun (Bronze) culture in the Korean Peninsula at ca. 3.4 ka BP, with clear evidence of intensive agriculture, including domesticated plants such as rice. The impact of temporary climate deterioration around 3.2 ka BP (Fig. 6b–d), which may reflect enhanced ENSO activity (Moy et al., 2002) (Fig. 6e) and/or the 3.2-ka event (Kaniewski et al., 2017), was

not large enough in the Korean Peninsula to interrupt the increasing population trend during this cultural boom (Fig. 6a).

During ca. 4.8–3.6 ka BP, high Ti values and a low proportion of arboreal pollen indicate a cool and dry climate in the Korean Peninsula when human activity declined (Fig. 6). Evidence of significant cooling and drying events around this period (Bond et al., 1997; Wang et al., 2005; Wanner et al., 2011; Bond et al., 2001) and their potential impacts on the

shrinkage or unrest of past societies (DeMenocal, 2001) have been widely reported from various sites worldwide including Mesopotamia (Weiss et al., 1993; Carolin et al., 2019), India (Dixit et al., 2018; Dixit et al., 2014), China (An et al., 2005; Li et al., 2018; Xiao et al., 2018; Xu et al., 2019), Japan (Kajita et al., 2020; Kawahata et al., 2009), and Korea (Constantine et al., 2019; Park et al., 2019). In these periods, climate drying would have increased dietary stress by hindering successful hunting, gathering, millet cultivation and even livestock domestication (Roffet-Salque et al., 2018; Kawahata et al., 2009).

Abrupt and synchronous changes in Ti and pollen data together with the decline in archaeological SPD values at ca. 4.8, 4.2, and 4.0 ka BP imply a significant impact of sudden climate deterioration on Korean prehistoric societies (Fig. 6). These changes were likely triggered by the onset of an active ENSO at ca. 4.8 ka BP (Fig. 6e) and modulated by lower SSTs in the Okinawa Trough and WTP until ca. 3.6 ka BP (Fig. 6f).

Our Ti XRF data suggest synchronicity of climate deterioration with a decline in population in the Korean Peninsula from ca. 2.8 ka BP until ca. 2.3 ka BP (Fig. 6a–d). During this period, SSTs in the Okinawa Trough and WTP decreased (Fig. 4f and g), and ENSO activity around 2.7 ka BP (Fig. 6e) may have amplified the climate impact. Although no palynological record is available for this period (Fig. 3 and 6c), our previous study of Gwangyang (Park et al., 2019) (Fig. 1b, GY-1) supports this socio-environmental link (Fig. 6d). In the Korean Peninsula, most Bronze Age pottery disappears during this period, as

observed for Misa-ri, Garak-dong, and Heunam-ri-type pottery around 2.9–2.8 ka BP and Yeoksam-dong, Songguk-ri, and Geomdan-ri-type pottery during 2.8–2.4 ka BP (Lee, 2017; Ahn and Hwang, 2015). In this sense, it is likely that decreasing SPD values after ca. 2.8 ka BP (Fig. 6a) primarily reflect a collapse of Bronze Age culture in Korea, and that this change was influenced by climate drying and/or cooling at that time (Fig. 6b-d). As during ca. 4.8–3.6 ka BP, the population decline after ca. 2.8 ka BP was a widespread phenomenon that has been detected in Korea and at many sites worldwide including

mainland China (Wang et al., 2014), Turkey (Woodbridge et al., 2019), and Britain and Ireland (Bevan et al., 2017). Therefore, climate impact on human societies during this period should be understood within a global context, possibly in connection with a cooling trend after the Bond event 2 (Bond et al., 1997; Wanner et al., 2011).



## 6 Conclusion

Our multi-proxy analysis of pollen, XRF, and grain-size data was used to reconstruct the EASM history of the Korean
Peninsula from ca. 8.3 to 2.3 ka BP. We identified a concurrent change between hydroclimate and past human activity. The
Holocene climate in Korea was sensitively modulated by the strength of the KC and ENSO-like variance. Wet conditions
prevailed during ca. 8.3–7.5, 7.1–6.4, 6.0–4.8, and 3.6–2.8 ka BP, when SSTs in the western Pacific Ocean were sufficiently
high to enhance EASM strength. However, during ca. 7.5–7.1, 6.4–6.0, and 4.8–3.6 ka BP, climate became drier due to KC
weakening and an increase in ENSO-like variance that likely dampened the EASM. Although regional imprints of climate
deterioration during ca. 6.4–6.0 ka BP remain unclear in East Asia, the findings of our study contribute to our knowledge of
this climate event in the Korean Peninsula. The reconstructed hydroclimate change was also synchronous with past
population levels inferred by the SPD of archaeological remains. Past societies flourished amid favorable climate conditions
during ca. 6–4.8 ka BP and ca. 3.6–2.8 ka BP, but suffered from precipitation deficits during ca. 4.8–3.6 ka BP. This finding
is consistent with multiple findings of contemporary collapses of past civilizations worldwide and confirms that this global
socio-environmental link was present in the Korean Peninsula. Nevertheless, it should be noted that the relationship between
climate and past societies is not always straightforward. Further research is required to elaborate our understanding of its
dynamics.

## Data availability

Data used in this study are available from Jinheum Park (jinheum94@snu.ac.kr) or Jungjae Park (jungjaep@snu.ac.kr) on
reasonable request. The data are also available at the open-access repository Pangaea.de (DOI not provided yet).

## Author contribution

Jinheum Park: Conceptualization, Formal analysis, Investigation, Writing – Original Draft, Visualization. Jungjae Park:
Conceptualization, Methodology, Writing – Review & Editing, Supervision, Project administration, Funding acquisition.
Sangheon Yi: Methodology, Resources, Project administration. Jaesoo Lim: Investigation. Jin Cheul Kim: Investigation.
Quihong Jin: Investigation. Jieun Choi: Investigation

## Competing interests

The authors declare that they have no conflict of interest.



**Acknowledgements**

This work was supported by the Basic Research Project (GP2017-013) of the Korea Institute of Geoscience and Mineral
Resources (KIGAM), the National Research Foundation of Korea (NRF-2018S1A2A01025813), and the Hanmaum Peace
and Research Foundation (HPRF). We thank the editor and anonymous reviewers for their useful comments and suggestions
for improving the manuscript.

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

**Figure 1: (a) Locations of the STP18-03 core (yellow star, this study) and other proxies mentioned in this article (white squares): A7 from Okinawa Trough (Sun et al., 2005), MD98-2176 from the Arafura Sea (Stott et al., 2004), Lake Xiaolongwan (Chu et al., 2014; Xu et al., 2019; Xu et al., 2014), Lake Sihailongwan (Stebich et al., 2015), Daihai Lake (Xiao et al., 2004), Gonghai Lake (Chen et al., 2015a), Qinghai Lake (Ji et al., 2005), and Dongge Cave (Wang et al., 2005; Dykoski et al., 2005). Red arrows indicate**
**the trajectory of the Kuroshio Current. Blue arrows indicate the trajectories of the East Asian Summer Monsoon (EASM) and Indian Summer Monsoon (ISM). (b) Map of the Korean Peninsula, showing the locations of GY-1 (Park et al., 2019), Pomaeho Lake (Constantine et al., 2019), and Jeju Island (Park, 2017). The maps in (a) and (b) were produced using the GMRT tool (Ryan et al., 2009). (c) Regional satellite map of our coring site (yellow star), showing the locations of the Jongnamsan (663 m) and Palbongsan (391 m) mountains and the Miryang River. This map was adapted from the © Google Earth Pro software ver.**
**7.3.3.7673 (https://earth.google.com/). (d) Mean monthly temperature and precipitation in Miryang during 1981–2010 (Korea Meteorological Administration, 2020).**





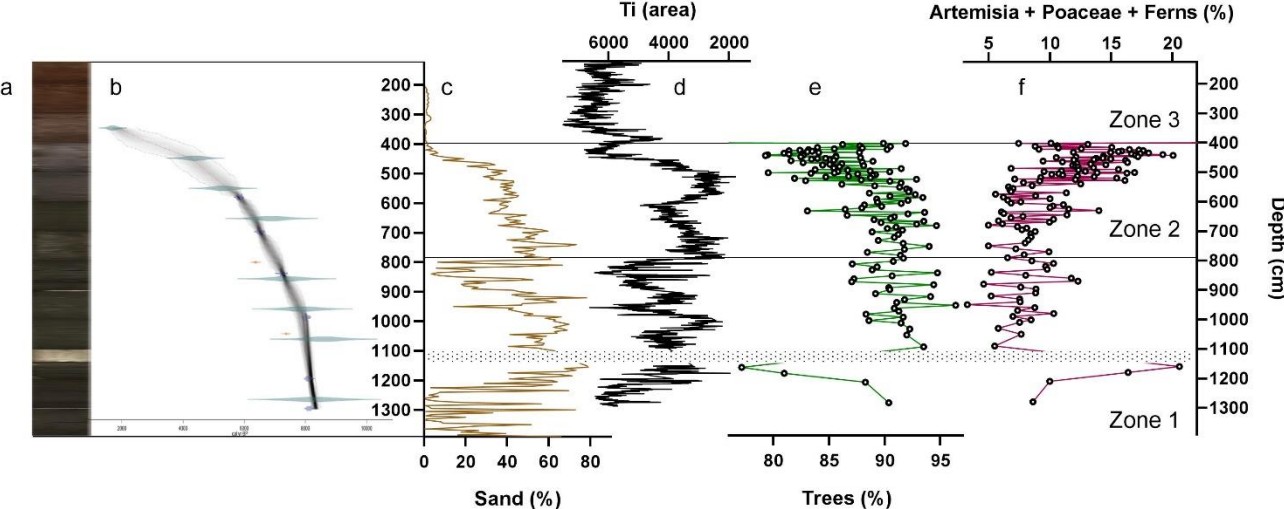

**Figure 2: (a) Digital image of the STP18-03 core and (b) age–depth model constructed using the R *bacon* package ver. 2.3 (Blaauw**
**and Christen, 2011) with the IntCal13 calibration dataset (Reimer et al., 2013). Samples omitted from the chronology model are**
**indicated in red. (c–f) results of multi-proxy analyses of (c) sand proportion, brown; (d) titanium (Ti) content, black; (e) tree pollen**
**percentage, green; and (f) sum of *Artemisia* (mugwort) and *Poaceae* (wild grass) pollen and fern spores, magenta. Zones are**
**separated by black horizontal lines.**



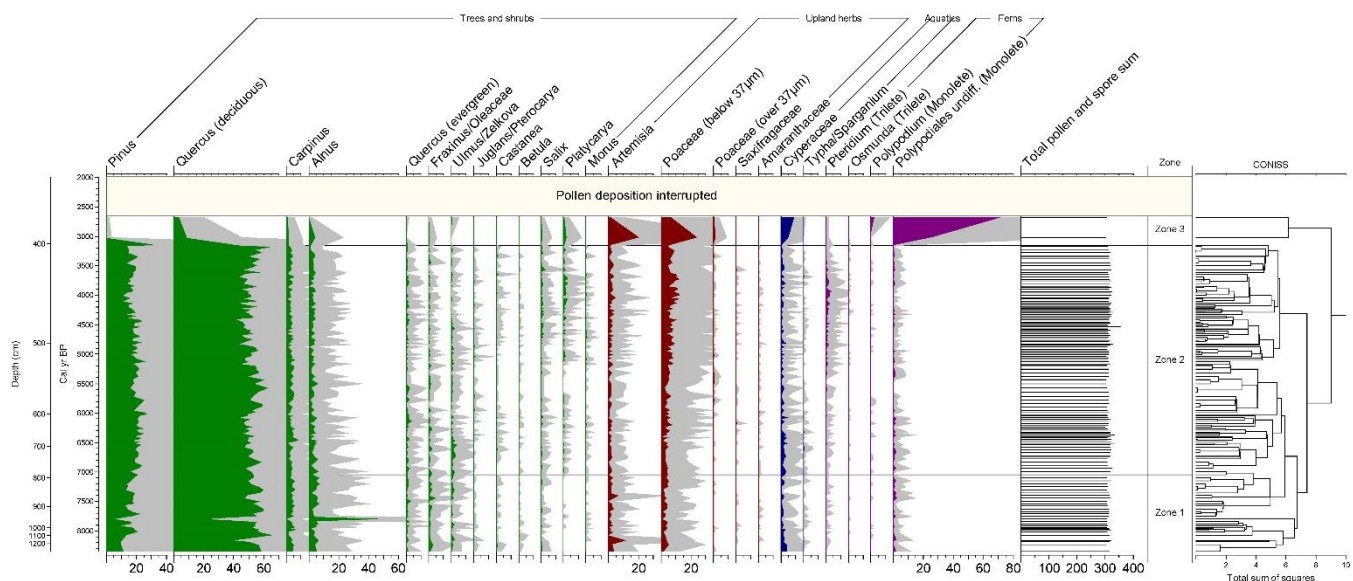


**Figure 3: STP18-03 pollen diagram of selected taxa. All pollen percentages were calculated based on the sum of all terrestrial pollen and spores.**


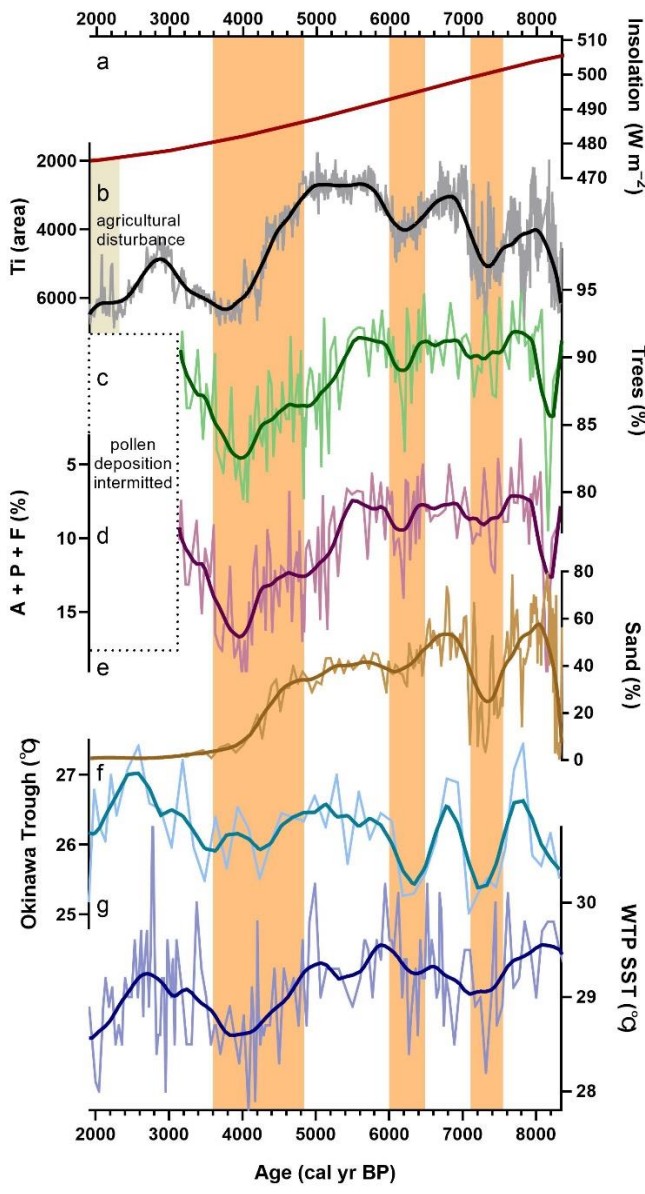

**Figure 4: Comparison of results for paleoclimate proxies. (a)** June insolation curve at 30°N, red (Berger and Loutre, 1991); **(b)** titanium content, black (this study); **(c)** tree pollen percentage in Miryang, green (this study); **(d)** sum of *Artemisia* (mugwort) and *Poaceae* (wild grass) pollen and fern spores, magenta (this study); **(e)** sand proportion over 63 μm in diameter, brown (this study); **(f)** sea surface temperature (SST) records from the A7 core, Okinawa Trough (Sun et al., 2005) (Fig. 1a), cyan; and **(g)** SST records from MD98-2176, Western Tropical Pacific (Stott et al., 2004) (Fig. 1a), blue. Smoothed lines in (b–g) were calculated using locally weighted scatterplot smoothing (LOWESS) regressions with 20-point windows. Vertical orange boxes indicate periods of drier climate. Box with black dotted edges indicates a zone with intermittent pollen deposition. Brown shading indicates an estimated period of agricultural disturbance following Yoon et al. (2005).




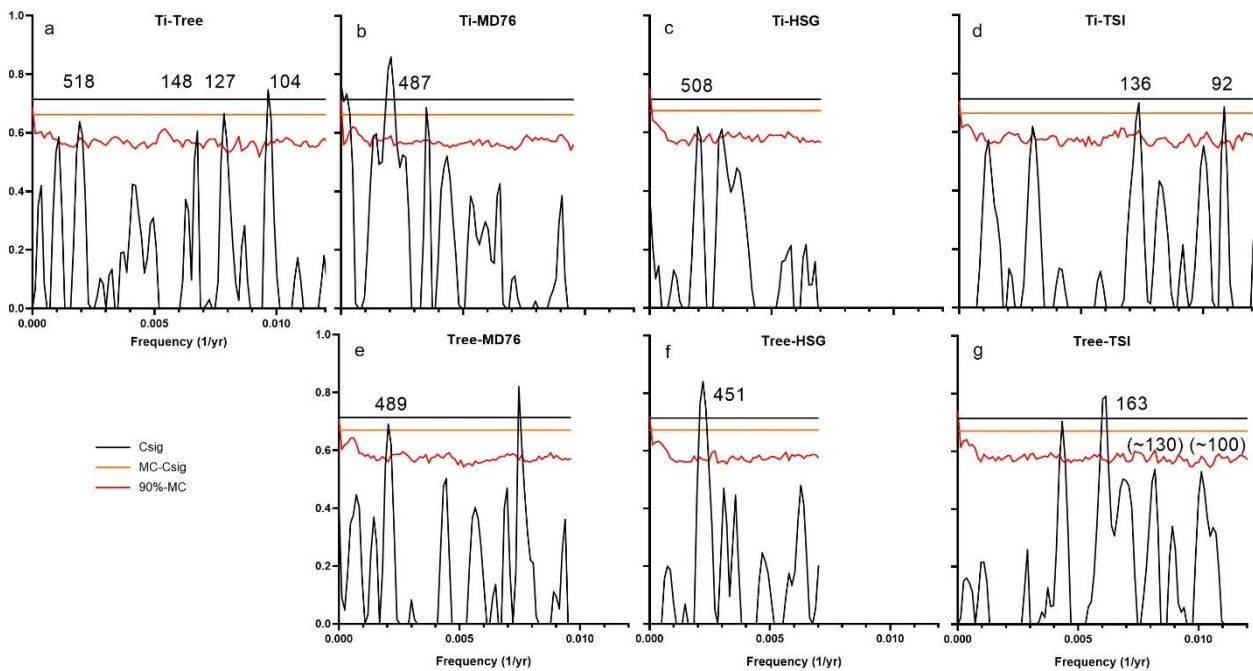

**Figure 5: Coherency spectra for (a) titanium – tree percentage, (b) titanium – MD76 (Stott et al., 2004), (c) titanium – hematite-stained grains (HSG) (Bond et al., 2001), (d) titanium – total solar irradiation (TSI) (Steinhilber et al., 2009), (e) tree – MD76, (f) tree – HSG, and (g) tree – TSI. Theoretical, mean Monte Carlo (for alpha = 0.050), and 90 % Monte Carlo false alarm levels are indicated by black, orange, and red lines, respectively. All cross-spectral analyses were conducted using the REDFIT-X software ver. 1.1 (Ólafsdóttir et al., 2016).**

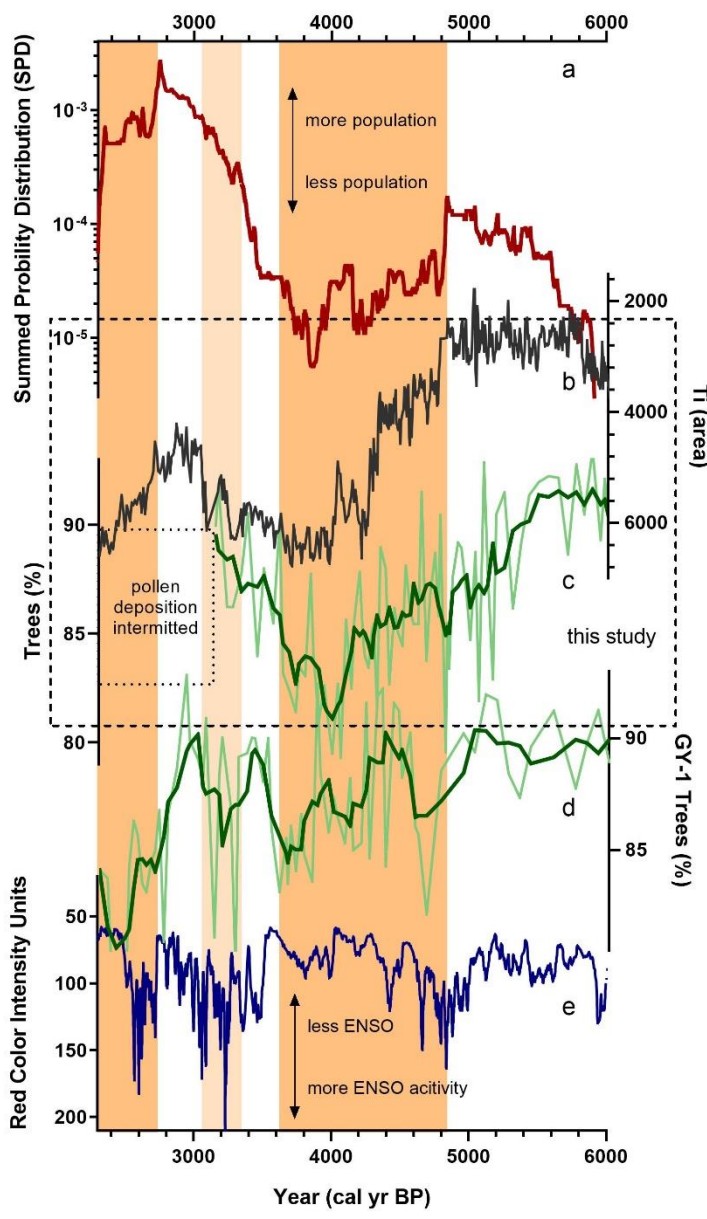

**Figure 6: Comparison of the results of paleoclimate proxies and past human activity indicators. (a) Normalized summed probability distribution (SPD) values on archaeological records found in the Korean Peninsula, red (Oh et al., 2017); (b) titanium content in STP18-03, black (this study); and (c) tree pollen percentages from STP18-03, green (this study) and (d) GY-1, green (Park et al., 2019) (Fig. 1b); and (e) red color intensity units from Laguna Pallcacocha, an indicator of El Niño–Southern Oscillation (ENSO), blue (Moy et al., 2002). Vertical orange boxes indicate periods of drier climate and lower population. Raw values were smoothed in (c) and (d) by 5-point means and in (e) by 31-point means.**





| OSL sample depth (m) | Dose rate (Gy/ka) | Water content (%) | Equivalent dose (Gy) | OSL age (yr BP) |
|---|---|---|---|---|
| 3.50 - 3.55 | 2.94 ± 0.20 | 28.3 | 5.09 ± 0.20 | 1700 ± 130 |
| 4.50 - 4.55 | 3.23 ± 0.20 | 37.4 | 14.09 ± 0.19 | 4400 ± 270 |
| 5.50 - 5.55 | 3.55 ± 0.20 | 26.8 | 18.75 ± 0.40 | 5300 ± 320 |
| 6.50 - 6.55 | 3.21 ± 0.18 | 33.3 | 22.14 ± 0.58 | 6900 ± 430 |
| 8.50 - 8.55 | 3.21 ± 0.20 | 39.1 | 23.88 ± 0.20 | 7400 ± 460 |
| 9.50 - 9.55 | 3.08 ± 0.17 | 40.0 | 24.32 ± 0.45 | 7900 ± 470 |
| 10.50 - 10.55 | 3.39 ± 0.19 | 30.6 | 29.28 ± 0.48 | 8600 ± 500 |
| 12.50 - 12.55 | 3.02 ± 0.19 | 40.0 | 25.25 ± 0.99 | 8300 ± 620 |
| AMS sample depth (m) | Material dated | Lab code | δ13C (‰) | Age ($^{14}$C yr BP) | Calibrated age (yr BP) |
| 5.80 | Plant fragments | KGM-ITg190725 | −28.8 | 5110 ± 40 | 5840 ± 90 |
| 6.94 | Plant fragments | KGM-ITg190726 | −27.7 | 5710 ± 40 | 6520 ± 110 |
| 7.95[a] | Plant fragments | KGM-ITg190727 | −32.5 | 5570 ± 50 | 6360 ± 80 |
| 8.32 | Plant fragments | KGM-ITg190728 | −32.0 | 6310 ± 40 | 7240 ± 80 |
| 9.77 | Plant fragments | KGM-ITg190729 | −33.9 | 7210 ± 40 | 8060 ± 100 |
| 10.32[a] | Plant fragments | KGM-ITg190730 | −30.8 | 6500 ± 50 | 7430 ± 120 |
| 11.81 | Plant fragments | KGM-ITg190731 | −30.8 | 7280 ± 40 | 8090 ± 80 |
| 12.80 | Plant fragments | KGM-ITg190732 | −27.1 | 7330 ± 50 | 8160 ± 140 |

620 **Table 1: Radiocarbon and optically stimulated luminescence (OSL) dating results for the STP18-03 core.**

**[a] Excluded from the age model.**



The English in this document has been checked by at least two professional editors, both native speakers of English. For a certificate, please see:

625

http://www.textcheck.com/certificate/PSeujQ