# Peer review of "Holocene hydroclimate reconstruction based on pollen, XRF, and grain-size analysis and its implications for past societies of the Korean Peninsula"

_Climate of the Past, 2020_

## Referee Comment (RC1) · Anonymous Referee #1 · 31 Aug 2020

The authors Park et al. used a sedimentary record from Miryang in the Korean Peninsula to describe climate induced hydrological changes for the Holocene period ca. 8.3-2.3 ka BP and indicate shifts in the human population with changing intensity of the East Asian Summer Monsoon (EASM). They mainly used a pollen record in combination with high-resolution titanium XRF scanning data and grain size variations to decipher the influence of the Kuroshio Current in the Pacific Ocean and El Nino Southern Oscillation (ENSO) in connection with solar forcing that influenced the development of EASM-regulated hydrologic variations. A summed probability distribution indicates

correspondence with changes in the local population in response to climate variations. This manuscript is well written, logically structured and provides reasonable explanations for the influences of different signals on hydrological variations and EASM impact. However, there are several aspects which need to be considered:

1. The chronological frame is not well explained. For example, there is no information how calendar ages related to OSL dating were connected with radiocarbon ages (uncalibrated) to develop a Bayesian age model (Blaauw and Christen, 2011). Furthermore, any explanation about potential reservoir errors is lacking. 2. With respect to the table1, it remains open, how the authors calibrated the 14C ages. Did they use 1 or 2 sigma uncertainties, and did they report mean, median or weighted mean values? How do these values differ from the Bacon age model? A column should be added to show this. The uncertainty values (in table 1) for the calibrated values are somewhat strange. They should check and correct it, while mentioning this in the results part under 4.1 Chronology. By the way, the age-depth model in figure 2b needs a readable age axis. It is impossible to read it because the axis description is far too small. Furthermore the authors should explain that the reported ages later in the discussion part refer to calibrated (cal.) ages BP or not. How did they deal with OSL calendar ages in this respect?

2. The explanation in the methods part is not sufficiently provided. How did they drill and how long were the core sequences? Furthermore how did they deal with potential sediment loss/overlapping at the boundary between the core segments? Finally how did they splice the different core sequences towards a composite one?

Furthermore, how was the instrumental setting for detecting titanium signals by XRF scans?

3. In chapter 4.2 the authors described the selected zones based on the provided data. This part is partly mixed with interpretation of data variations. The authors should perhaps change the title of chapter 4 (Results) to Results and interpretation. Furthermore,

I wonder why the authors did not provide graphs for the clay and silt fractions in addition to the sand fractions. In line 149 they mention that the sediments mainly consist of clay. This would we worthy to demonstrate this by the clay and silt fraction graphs. They could be attached to figure 2a.

4. Lines 229-234: The authors refer to a cooling trend around 6.4-6.0 ka BP and mention that this is not seen in other records. They used an example (lines 232-234), but to my understanding this explanation supports their finding. So, what are the differences? This part of the discussion remains not very clear and shall be considered for revision.

5. All parts of the discussion strongly rely on their provided chronology. Hence it is important to explain in more detail whether this chronology is reliable (see comment no.1).

---

## Referee Comment (RC2) · Anonymous Referee #2 · 1 Sep 2020

1. General comments: The authors present multiple geochemical proxies data from a floodplain sedimentary core in Miryang, Korea. High resolution pollen, gran size, and XRF-scanning data show clear oscillation during the past ∼8 kyrs.

Although authors would like to reconstruct the paleoclimate conditions in this region and to discuss the climatic controlling factors and how human societies response to climate changes, there are some questions need to be clarified.

2. Questions need to be clarified: Line 40, authors should clarify why the methodology

of pollen studies could not increase temporal resolution and what kind of resolution they would like to achieve (1-yr? 10-yr?). Temporal resolution is a relative concept and there are other factors need to be considered. If there has strong mixing or bioturbation, then even we could use few micrometer resolution to scan sediment core with XRF-scanner, it is still meaningless to claim the data we get has higher temporal resolution.

Line 49, author mentioned the relationship between climate change and human societies, but to specific, which part of human societies? Do authors mean the resilience of human societies, the limitation factors from climate changes, the total population changes, the food availability?

Line 73, do authors suggest we could explain the whole Korean Peninsula climate change with a single site and the whole peninsula population changes as well?

Line 77, authors mentioned this sedimentary core is derived from a floodplain, but authors did not provide clear evidence how sediments had been transported to here (by occasional flood? Or the main path of river had changed many times through time? Or it is a terrace that had been uplifted?). This is very important especially when authors would like to explain the grain size change to reconstruct climate changes in the past. If we don't know how sediment had been transported to here, then it is very dangerous to treat authors pollen and XRF-scanning as an in-situ signals.

Line 91, authors described modern forest species, but is it possible that author authors could provide surface soil pollen as a modern control to confirm that the modern assemblage of vegetation is similar to the pollen composition as well?

Line 100-103, why authors could identify the uppermost and the lowermost of sediment core are affected by human activities? Please provide clear evidence (pictures?) and explain why there is no human activities during their study period.

Line 109-110, why these two 14C dates are omitted? Why these 2 dates have anomalous ages? Please explain.

Line 160, similar to line 77, where does the Ti come from? In Lines 185-188, authors claim they interpret Ti signal as a reflection change in the "Korean Peninsula". But there are no references to support their interpretations.

Furthermore, Ti does not "follow" the insolation changes, at least by my naked eyes. Authors could argue there is a clear increasing ~4.8 to 3.8-kyr, but the variabilities during 8-5 kyr show rather centennial oscillation than gradual shifting.

Finally, in lines 229-237, authors would like to connect their records to broad regional forcings, such as ITCZ, ENSO, and Kuroshio strength, however, they could not provide good interpretation to explain the differences between their records to other records with in the peninsula.

---

## Editor Comment (EC1) · Pierre Francus (Editor) · 7 Sep 2020

Dear authors,

I read the reviewers comments and also read your manuscript.

I agree with the reviewers that the sedimentological context is poorly explained and should be expended to demonstrate that you have a continuous record, and that there is no hiatus in the sedimentary sequence that you analysed. Reliability of the chronology should be demonstrated, and the sedimentological log should be improved.

[Figure]

While I'm not a palynologist, I'm also quite surprised that you have not attempted to apply a pollen transfer function to provide quantified hydroclimate reconstruction. Is there a justification for that?

I'm looking forward to reading your answer

Pierre Francus

---

## Author Comment (AC1) · 17 Sep 2020

Response to Anonymous Referee #1

We thank you very much for your insightful review. Your comments are highly appreciated.

We added our response below each of your comment.

The authors Park et al. used a sedimentary record from Miryang in the Korean Peninsula to describe climate induced hydrological changes for the Holocene period ca. 8.3-2.3 ka

BP and indicate shifts in the human population with changing intensity of the East Asian

Summer Monsoon (EASM). They mainly used a pollen record in combination with high- resolution titanium XRF scanning data and grain size variations to decipher the influence of the Kuroshio Current in the Pacific Ocean and El Nino Southern Oscillation (ENSO) in connection with solar forcing that influenced the development of EASM-regulated hydrologic variations. A summed probability distribution indicates correspondence with changes in the local population in response to climate variations. This manuscript is well written, logically structured and provides reasonable explanations for the influences of different signals on hydrological variations and EASM impact. However, there are several aspects which need to be considered:

1. The chronological frame is not well explained. For example, there is no information how calendar ages related to OSL dating were connected with radiocarbon ages (uncalibrated)

to develop a Bayesian age model (Blaauw and Christen, 2011).

Response: We modified the sentence in Lines 107–109 as follows: "Compiling the OSL and radiocarbon dating results, we constructed an age model using the *bacon* R package (Blaauw and Christen, 2011) ver. 2.3 (Fig. 2b). The package allows a combination of different types of dates in a single age-depth modelling. Here, the radiocarbon dates were calibrated based on the

IntCal13 calibration dataset (Reimer et al., 2013), while the OSL dates were not because they were already set on the calendar scale. All resulting ages applied in our analysis were expressed as calendar ages".

Furthermore, any explanation about potential reservoir errors is lacking.

Response: Essentially, the OSL dating method does not require a reservoir correction. For radiocarbon dates, general coherency with other OSL dating results (Fig. 2b) indicates that reservoir effect is negligible in our STP18-03 core.

2. With respect to the table1, it remains open, how the authors calibrated the 14C ages. Did they use 1 or 2 sigma uncertainties, and did they report mean, median or weighted mean values? How do these values differ from the Bacon age model? A column should be added to show this. The uncertainty values (in table 1) for the calibrated values are somewhat strange. They should check and correct it, while mentioning this in the results part under 4.1 Chronology.

Response: We replaced the calibrated ages in Table 1 to weighted mean ages as calculated from the *bacon* package (Blaauw and Christen, 2011) based on the IntCal13 dataset (Reimer et al., 2013), which are the same as used in our analysis. We added this information to the caption. The previous dates in Table 1 were the ones preliminarily calculated at the dating institution with OxCal (Ramsey, 1995), and they were not directly related to our age-depth model. We appreciate your noticing.

By the way, the age-depth model in figure 2b needs a readable age axis. It is impossible to read it because the axis description is far too small.

Response: We modified Fig. 2b as your direction.

Furthermore the authors should explain that the reported ages later in the discussion part refer to calibrated (cal.) ages BP or not. How did they deal with OSL calendar ages in this respect?

Response: We modified the sentence in Line 107–109 as follows: "Compiling the OSL and radiocarbon dating results, we constructed an age model using the *bacon* R package (Blaauw and Christen, 2011) ver. 2.3 (Fig. 2b). The package allows a combination of different types of dates in a single age-depth modelling. Here, the radiocarbon dates were calibrated based on the IntCal13 calibration dataset (Reimer et al., 2013), while the OSL dates were not because they were already on the calendar scale. All resulting ages applied in our analysis were expressed as
calendar ages".

2. The explanation in the methods part is not sufficiently provided. How did they drill and
   how long were the core sequences?

Response: The length of the core is 20 meters, as mentioned in Line 100. For drilling, we used
a hydraulic piston corer mounted on a truck. We modified the sentence in Line 100 as follows:
"In April 2018, the 20-m STP18-03 core was collected in 1-m sections from a former floodplain
of the Miryang River, using a hydraulic piston corer (Fig. 1)".

Furthermore how did they deal with potential sediment loss/overlapping at the boundary
   between the core segments? Finally how did they splice the different core sequences
   towards a composite one?

Response: Our hydraulic piston corer was mounted on a truck and drilled under stable
conditions. The truck was anchored on the solid ground while the drill was put into the hole
with consistent mechanic settings. Therefore, we assumed that potential loss or overlapping
between core the segments was minimal and spliced them without additional correction process.

Furthermore, how was the instrumental setting for detecting titanium signals by XRF scans?

Response: We separated the sentences in Line 127–133 as a new paragraph for better readability.

3. In chapter 4.2 the authors described the selected zones based on the provided data. This
   part is partly mixed with interpretation of data variations. The authors should perhaps
   change the title of chapter 4 (Results) to Results and interpretation.

Response: We modified the title of Chapter 4 as your advice: "Results and interpretation".

Furthermore, I wonder why the authors did not provide graphs for the clay and silt fractions
   in addition to the sand fractions. In line 149 they mention that the sediments mainly consist of clay. This would we worthy to demonstrate this by the clay and silt fraction graphs. They could be attached to figure 2a.

Response: We added the clay and silt fractions to Fig. 2 along with the sand fraction data. We also added a description of core lithology and changed the figure caption as follows: "Figure

2: (a) Lithology of the STP18-03 core and (b) ~~ (c–h) results of multi-proxy analyses of (c)

clay fraction, gray; (d) silt fraction, light brown; (e) sand fraction, dark brown; (f) titanium (Ti)

content, black; (g) tree pollen percentage, green; and (h) sum of Artemisia (mugwort) and

Poaceae (wild grass) pollen and fern spores, magenta. Zones are separated by black horizontal lines.".

  We also edited figure numbers throughout the text which are associated with Fig. 2.

  We modified the Line 149 as follows: "This zone consists mainly of very dark brown silt and sand alternating in multiple layers with ~15 % of clay (Fig. 2a and c–e)".

 We modified the Line 164 as follows: "From 790 to 400 cm, the clay and silt content gradually increase as depth decreases, from ~15 and ~30 % to ~20 to ~70 %, respectively (Fig.

2c and da).".

4. Lines 229-234: The authors refer to a cooling trend around 6.4-6.0 ka BP and mention

 that this is not seen in other records. They used an example (lines 232-234), but to my

 understanding this explanation supports their finding. So, what are the differences? This

 part of the discussion remains not very clear and shall be considered for revision.

Response: To clarify the discussion, we revised the paragraph (Lines 229–249) as follows:

"Among these periods, a sign of drying and/or cooling around 6.4–6.0 ka BP (Fig. 4b–e) at

Miryang is consistent with our previous finding at Lake Pomaeho in the central Korean

Peninsula (Constantine et al., 2019) (Fig. 1b). Outside of the peninsula, Daihai Lake (Xiao et al., 2004) and Gonghai Lake (Chen et al., 2015a) in North China and Dongge Cave in South

China (Wang et al., 2005) (Fig. 1a) also record abrupt shifts toward less precipitation at ca.

6.4–6.0 and 7.5–7.1 ka BP. These findings altogether suggest a possibility that the climate events were widespread phenomena in the East Asian region. Nevertheless, this possibility should be carefully addressed, as some study sites such as Lake Xiaolongwan (Chu et al., 2014;

Xu et al., 2019) and Lake Sihailongwan (Stebich et al., 2015) (Fig. 1a) do not clearly exhibit a drying/cooling signal. Regarding this inconsistency, a couple of possibilities can be considered. One possible factor is an issue of temporal resolution. In the case of Dongge Cave, the high-resolution DA stalagmite (Wang et al., 2005) detects a drying signal while the D4 stalagmite (Dykoski et al., 2005), with a lower resolution, does not. It is not reasonable to assume difference in actual climate conditions because they were collected from the same cave. Similarly, in the Korean Peninsula, our previous study at Gwangyang (Fig. 1b, GY-1) does not exhibit a climate shift at ca. 6.4–6.0 ka BP (Park et al., 2019) in contrast to Miryang (this study). As Gwangyang is located only ~100 km west to Miryang, it is unlikely that climate conditions were considerably different between those two study sites. Rather, temporal resolution is a more convincing explanation as the sample intervals covering the period are large in GY-1 (~80 years) relative to our present study (~20–30 years).

Besides the resolution issue, potential bias inherent in proxy-based climate reconstructions should be also noted. In pollen records, source area and/or overestimation effects inherent in palynological methodology (Seppä and Bennett, 2003) might affect pure climate signals. For example, in this study, we suspect that thermal optimum during the early to mid-Holocene (Wanner et al., 2008) might have rendered the smaller amplitude of the vegetation response during ca. 7.5–7.1 ka BP, whereas the other sedimentary proxies, XRF and sand percentage data exhibit clearer phase shifts with the Pacific Ocean (Fig. 4b–g). Similarly, in pollen records from Daihai Lake (Xiao et al., 2004) and Gonghai Lake (Chen et al., 2015a), drying signals during ca. 7.5–7.1 ka BP are less evident than ca. 6.4–6.0 ka BP. In this context, it cannot be ruled out that such climate shifts are not manifest in some records simply due to methodological problems. Furthermore, limiting to the cases of Lake Xiaolongwan (Chu et al., 2014; Xu et al., 2019) and Lake Sihailongwan (Stebich et al., 2015) in Northeast China (Fig. 1a), regionally varying climate imprints caused by high-latitude forcing such as sea ice in the Sea of Okhotsk (Stebich et al., 2015) should also be considered although this is beyond our research scope. Overall, in order to elaborate understanding on potential climate deterioration events at ca. 6.4–6.0 and 7.5–7.1 ka BP, further high-resolution data are required from multiple locations in East Asia. At least in this study, our finding at Miryang adds to evidence that such climate shifts were likely present in the Korean Peninsula during these two periods".

5. All parts of the discussion strongly rely on their provided chronology. Hence it is important to explain in more detail whether this chronology is reliable (see comment no.1).

Response: We added a sentence in Sect. 4.1 as follows: "Throughout the age-depth model, our
dating results exhibited high coherency despite two different methodologies used, OSL and
radiocarbon dating (Fig. 2b)".

For details regarding construction of the age-depth model, please refer to our responses
above.

                                  References

Blaauw, M. & Christen, J. A. 2011. Flexible paleoclimate age-depth models using an
autoregressive gamma process. *Bayesian analysis,* 6**,** 457-474.
Ramsey, C. B. 1995. Radiocarbon calibration and analysis of stratigraphy: the OxCal program.
*Radiocarbon,* 37**,** 425-430.
Reimer, P. J., Bard, E., Bayliss, A., Beck, J. W., Blackwell, P. G., Ramsey, C. B., Buck, C. E.,
Cheng, H., Edwards, R. L. & Friedrich, M. 2013. IntCal13 and Marine13 radiocarbon
age calibration curves 0–50,000 years cal BP. *Radiocarbon,* 55**,** 1869-1887.

[Figure]

**Figure 2: (a)** Lithology **of the STP18-03 core and (b) age–depth model constructed using the R** *bacon* **package ver. 2.3 (Blaauw and Christen, 2011) with the IntCal13 calibration dataset (Reimer et al., 2013). Samples omitted from the chronology model are indicated in red.** (c–h) results of multi-proxy analyses of (c) clay fraction, gray; (d) silt fraction, light brown; (e) sand fraction, dark brown; (f) titanium (Ti) content, black; (g) tree pollen percentage, green; and (h) sum of *Artemisia* (mugwort) and *Poaceae* (wild grass) pollen and fern spores, magenta. Zones are separated by black horizontal lines.

---

## Author Comment (AC2) · 17 Sep 2020

Response to Anonymous Referee #2

We highly appreciate your constructive and insightful comments. Please refer to our response
written below each of your comment.

1. General comments: The authors present multiple geochemical proxies data from a
floodplain sedimentary core in Miryang, Korea. High resolution pollen, gran size, and
XRF-scanning data show clear oscillation during the past ~8 kyrs. Although authors would
like to reconstruct the paleoclimate conditions in this region and to discuss the climatic
controlling factors and how human societies response to climate changes, there are some
questions need to be clarified.

2. Questions need to be clarified: Line 40, authors should clarify why the methodology of
pollen studies could not increase temporal resolution and what kind of resolution they
would like to achieve (1-yr? 10-yr?). Temporal resolution is a relative concept and there
are other factors need to be considered. If there has strong mixing or bioturbation, then
even we could use few micrometer resolution to scan sediment core with XRF-scanner, it
is still meaningless to claim the data we get has higher temporal resolution.

Response: We modified Lines 41–43 as follows: "~~ this methodology contains physical
limitations in enhancing the temporal resolution between samples (here, we refer to a temporal
resolution issue in a relative sense to other methodologies when assuming equal reliability in
sedimentary conditions). This problem arises from the fact that a certain amount (usually ~1 g)
of bulk sediment should be manually picked for pollen sample preparation, while some other
methodologies are available by drilling a smaller amount of sample (Lachniet, 2009) or even
mechanically scanning sediment surfaces (Croudace et al., 2006). On the other hand,
speleothem-based Holocene EASM reconstructions in the Korean Peninsula, which usually
permit finer resolution than pollen analysis, have been confined to the last few thousand years
and have not ~~"

Line 49, author mentioned the relationship between climate change and human societies, but to specific, which part of human societies? Do authors mean the resilience of human societies, the limitation factors from climate changes, the total population changes, the food availability?

Response: We elaborated the sentence as follows: "~~ between climate change and human societies, including issues of population/migration (D'Andrea et al., 2011; Tallavaara et al., 2015), rise and fall of civilizations (Weiss et al., 1993), and societal unrest (Manning et al., 2017)".

Line 73, do authors suggest we could explain the whole Korean Peninsula climate change with a single site and the whole peninsula population changes as well?

Response: As a subregion of East Asia, the Korean Peninsula is a relatively small area in a regional sense. Validity of using the core STP18-03 was also supported by providing another pollen record (GY-1) reconstructed within the peninsula (Fig. 6d), which showed a close resemblance to Miryang in this study (Fig. 6c). For more details on the GY-1 core, please refer to Park et al. (2019). Although there is some discrepancy between the two records during ca. 6.4–6.0 ka BP, this issue was discussed with regards to a problem of temporal resolution in Sect. 5.2 (Please refer to a revised paragraph provided at the end of this document).

Line 77, authors mentioned this sedimentary core is derived from a floodplain, but authors did not provide clear evidence how sediments had been transported to here (by occasional flood? Or the main path of river had changed many times through time? Or it is a terrace that had been uplifted?). This is very important especially when authors would like to explain the grain size change to reconstruct climate changes in the past. If we don't know how sediment had been transported to here, then it is very dangerous to treat authors pollen and XRF-scanning as an in-situ signals.

Response: First, we should clarify that the site has now been reclaimed into a rice paddy. Accordingly, we modified sentences in Lines 77–78 as " was collected in 1-m sections from

As the sedimentation process on the site cannot be deduced from modern observations,
we dedicated the first half of Sect 5.1 to discussing potential sediment transport mechanism to
the coring site during the Holocene. Considering the regional setting in which the coring site
is backed by small mountains while facing a small tributary (Miryang River) in front (Fig. 1c),
we interpreted the sediments to have been accumulated by relative amounts of soils eroded
from the background mountains as well as those transported by river discharges. This
interpretation was based on consistently opposing trends of the Ti and sand content in our
analysis and further supported by general coherency with the tree pollen percentages (Fig. 4b–
e). During wet conditions, sand contribution by river discharge would have increased, while
soil erosion from the background mountains would have decreased owing to enhanced tree
growth and anchoring effect of tree roots, and vice versa during dry periods (Please see Sect.
5.1). If they were not effectively reflecting hydroclimate change, they would not have achieved
such a close relationship with each other throughout such a long period.

As the coring location was formerly a floodplain, occasional floods would certainly
have happened in the area. However, there is no trace of large debris throughout the core, while
core chronology is stable (Fig. 2). Therefore, it seems likely that our coring location was far
enough from the river to be free of direct disturbance and that hydroclimatic implications of
our multi-proxy data are reliable. The sedimentary environment of the coring location would
have been sustained stable while being only modestly sourced by river discharge reflected as
the sand fraction.

Likewise, a possibility of direct disturbance effect by the river path is extremely low,
considering the stability in the age-depth model, absence of large debris as well as persistence
of dark brown sediment color throughout the core (only except the reddish-brown one on the
core top, which we interpreted as disturbed by human activity) (Fig. 2). In this context, it is
likely that the coring site persisted as a stable floodplain during the Holocene before eventually
desiccating at ca. 3.1 ka BP, which is implied by a near absence of the sand content and
intermittence of pollen deposition (Fig. 4b and c). A possibility of terrace/uplift can also be
obviated, as the sediment pertains to the Holocene period only, not a glacial-interglacial
transition. There is not any geological evidence to support the terrace/uplift hypothesis.

We elaborated the sentence in Lines 188–190 as follows: "~~ in nearby mountains such as Jongnamsan and Palbongsan (Fig. 1c) hills would have suppressed soil erosion via the anchoring effect of roots, leading to lower Ti XRF values".

For clarification of the meaning, we also modified Line 176 as follows: "Considering the near absence of sand content in this zone and coincident change in the sediment colour (Fig. 2a and e), this disruption may have been caused by a cessation of water supply from the river to the floodplain, possibly due to climate drying, dwindling of the river, and subsequent exposure of the site to air".

Line 91, authors described modern forest species, but is it possible that author authors could provide surface soil pollen as a modern control to confirm that the modern assemblage of vegetation is similar to the pollen composition as well?

Response: We appreciate your comment but unfortunately, it is not available. Since a retrieved rice paddy now lies on the location, it is unlikely that the surface pollen would reflect the nearby vegetation as it had done during the Holocene as a wetland state.

Line 100-103, why authors could identify the uppermost and the lowermost of sediment core are affected by human activities? Please provide clear evidence (pictures?) and explain why there is no human activities during their study period.

Response: In the text, we attributed the uppermost part only (not the lowermost one) to human disturbance (please see Lines 100–102). For the uppermost part (0–125 cm), such interpretation was made because the location is now used as a rice paddy. This information is now clarified in the text by modifying sentences in Lines 77–78 as "~~ located in a former floodplain of the Miryang River in the southeastern part of the Korean Peninsula (Fig. 1a–c), which is now reclaimed as a rice paddy". Also, we modified the sentence in Lines 101–102 as follows: "as the former were regarded to have been disturbed by agriculture and the latter consisted mainly of gravel". Also, it should be noted that our study period is a result of an even more conservative approach regarding the issue of human activities. Referring to Yoon et al. (2005) which detected evidence of agriculture in the Miryang area from ca. 2.3 ka BP, we refrained from making any paleoenvironmental interpretation on our data above 365 cm (not just above 125 cm), which corresponds to the time of ca. 2.3 ka BP.

Line 109-110, why these two 14C dates are omitted? Why these 2 dates have anomalous ages? Please explain.

Response: Except those two radiocarbon dates (795 and 1032 cm), dating results exhibited high coherency along the core despite two different methodologies used (OSL and radiocarbon). However, only these two radiocarbon dates among a total of 16 dates were anomalous and therefore excluded from the age-depth model. I hope for your understanding that in age-depth modelling of paleo-studies, it is not always possible to figure out exact reasons for every anomalous dating result. In this study, we might think of potential disturbance effect by plant roots, for example. One may argue a possibility that these two dates are actually correct and that the other (generally older) six radiocarbon dates are artefacts of reservoir effect, which is a common source of error when dealing with radiocarbon dating results. However, at least in our model this hypothesis can be obviated, as these six dates are highly coherent with the nearby OSL dates which are free of reservoir effect.

Line 160, similar to line 77, where does the Ti come from?

Response: We interpreted Ti as soil eroded from mountains in the background such as Jongnamsan and Palbongsan (Fig. 1c). For details, please refer to our response above.

In Lines 185-188, authors claim they interpret Ti signal as a reflection change in the "Korean Peninsula". But there are no references to support their interpretations.

Response: We modified the expression as follows: "~~ as reflecting hydroclimate change in the study area".

Furthermore, Ti does not "follow" the insolation changes, at least by my naked eyes. Authors could argue there is a clear increasing ~4.8 to 3.8-kyr, but the variabilities during 8-5 kyr show rather centennial oscillation than gradual shifting.

Response: We modified Line 179 as follows: "". We also modified Lines 184–185 as follows: "While centennial oscillations are relatively pronounced during the early–mid Holocene, Ti XRF values also change in tandem with these two proxies such that the signal generally increases towards the late Holocene (Fig. 4b)".

Finally, in lines 229-237, authors would like to connect their records to broad regional forcings, such as ITCZ, ENSO, and Kuroshio strength, however, they could not provide good interpretation to explain the differences between their records to other records with in the peninsula.

Response: To clarify the discussion, we revised the paragraph (Lines 229–249) as follows: "Among these periods, a sign of drying and/or cooling around 6.4–6.0 ka BP (Fig. 4b–e) at Miryang is consistent with our previous finding at Lake Pomaeho in the central Korean Peninsula (Constantine et al., 2019) (Fig. 1b). Outside of the peninsula, Daihai Lake (Xiao et al., 2004) and Gonghai Lake (Chen et al., 2015a) in North China and Dongge Cave in South China (Wang et al., 2005) (Fig. 1a) also record abrupt shifts toward less precipitation at ca. 6.4–6.0 and 7.5–7.1 ka BP. These findings altogether suggest a possibility that the climate events were widespread phenomena in the East Asian region. Nevertheless, this possibility should be carefully addressed, as some study sites such as Lake Xiaolongwan (Chu et al., 2014; Xu et al., 2019) and Lake Sihailongwan (Stebich et al., 2015) (Fig. 1a) do not clearly exhibit a drying/cooling signal. Regarding this inconsistency, a couple of possibilities can be considered. One possible factor is an issue of temporal resolution. In the case of Dongge Cave, the high-resolution DA stalagmite (Wang et al., 2005) detects a drying signal while the D4 stalagmite (Dykoski et al., 2005), with a lower resolution, does not. It is not reasonable to assume difference in actual climate conditions because they were collected from the same cave. Similarly, in the Korean Peninsula, our previous study at Gwangyang (Fig. 1b, GY-1) does not exhibit a climate shift at ca. 6.4–6.0 ka BP (Park et al., 2019) in contrast to Miryang (this study). As Gwangyang is located only ~100 km west to Miryang, it is unlikely that climate conditions were considerably different between those two study sites. Rather, temporal resolution is a more convincing explanation as the sample intervals covering the period are large in GY-1 (~80 years) relative to our present study (~20–30 years).

Besides the resolution issue, potential bias inherent in proxy-based climate reconstructions should be also noted. In pollen records, source area and/or overestimation effects inherent in palynological methodology (Seppä and Bennett, 2003) might affect pure climate signals. For example, in this study, we suspect that thermal optimum during the early to mid-Holocene (Wanner et al., 2008) might have rendered the smaller amplitude of the vegetation response during ca. 7.5–7.1 ka BP, whereas the other sedimentary proxies, XRF and sand percentage data exhibit clearer phase shifts with the Pacific Ocean (Fig. 4b–g). Similarly, in pollen records from Daihai Lake (Xiao et al., 2004) and Gonghai Lake (Chen et al., 2015a), drying signals during ca. 7.5–7.1 ka BP are less evident than ca. 6.4–6.0 ka BP. In this context, it cannot be ruled out that such climate shifts are not manifest in some records simply due to methodological problems. Furthermore, limiting to the cases of Lake Xiaolongwan (Chu et al., 2014; Xu et al., 2019) and Lake Sihailongwan (Stebich et al., 2015) in Northeast China (Fig. 1a), regionally varying climate imprints caused by high-latitude forcing such as sea ice in the Sea of Okhotsk (Stebich et al., 2015) should also be considered although this is beyond our research scope. Overall, in order to elaborate understanding on potential climate deterioration events at ca. 6.4–6.0 and 7.5–7.1 ka BP, further high-resolution data are required from multiple locations in East Asia. At least in this study, our finding at Miryang adds to evidence that such climate shifts were likely present in the Korean Peninsula during these two periods".

References

Park, J., Park, J., Yi, S., Kim, J. C., Lee, E. & Choi, J. 2019. Abrupt Holocene climate shifts in coastal East Asia, including the 8.2 ka, 4.2 ka, and 2.8 ka BP events, and societal responses on the Korean peninsula. *Scientific reports,* 9**,** 1-16.

[Figure]

**Figure 2: (a)** Lithology **of the STP18-03 core and (b) age–depth model constructed using the R** *bacon* **package ver. 2.3 (Blaauw and Christen, 2011) with the IntCal13 calibration dataset (Reimer et al., 2013). Samples omitted from the chronology model are indicated in red.** **(c–h) results of multi-proxy analyses of (c) clay fraction, gray; (d) silt fraction, light brown; (e) sand fraction, dark brown; (f) titanium (Ti) content, black; (g) tree pollen percentage, green; and (h) sum of** *Artemisia* **(mugwort) and** *Poaceae* **(wild grass) pollen and fern spores, magenta. Zones are separated by black horizontal lines.**

---

## Author Comment (AC3) · 17 Sep 2020

Response to Editor Comment 1

Dear Editor,

We thank you very much for your insightful comments.

Regarding the issues of chronology and sedimentology, please refer to our responses to the reviewers uploaded above.

We admit that they were poorly demonstrated in the original text (preprint) and tried to improve them by modifying/adding a significant amount of relevant descriptions.

Regarding the issue of quantitative climate reconstruction, we are sorry that it was not available due to scarcity of relevant base data in the Korean Peninsula.

We acknowledge that there are abundant datasets constructed in China. However, we did not use them because we could not guarantee their applicability to our pollen record acquired in the Korean Peninsula, which is a geographically different sub-region of East Asia.

Nevertheless, as your insightful comment, we agree that quantitative methods would greatly contribute to advances in paleoclimate studies and hope that they become widely available in the Korean Peninsula as well in the near future.

Please feel free to let us know anytime if there is any further point to be discussed or replied.

Sincerely Yours,

Jinheum Park (on behalf of authors)